# Co-Promoted CoNi Bimetallic Nanocatalyst for the Highly Efficient Catalytic Hydrogenation of Olefins

**DOI:** 10.3390/nano13131939

**Published:** 2023-06-26

**Authors:** Fei Wu, Yueying Wang, Shunxin Fei, Gang Zhu

**Affiliations:** 1Wuhan Institute of Marine Electric Propulsion, Wuhan 430064, China; 2School of Materials Science & Engineering, Anhui University of Technology, Maanshan 243002, China

**Keywords:** CoNi bimetal, nanoparticles, catalytic hydrogenation, promotion effect, charge transfer

## Abstract

Bimetallic catalysts, especially non-noble metals, hold great potential for substituting for noble metals in catalytic hydrogenation. In present study, a series of Co*_x_*Ni*_y_* (*x* + *y* = 6) bimetallic catalysts were prepared through the impregnation–reduction method and cyclohexene was chosen as probe-molecule to study the promotion effect of Co on the catalytic olefin hydrogenation reactions. Meanwhile, density functional theory (DFT) was utilized to investigate the formation energies and the charge distribution of CoNi bimetals, as well as the transition state (TS) searches for hydrogen dissociation and migration. The results suggest that bimetals tend to have superior catalytic performance than pure metals, and Co_3_Ni_3_ shows the highest catalytic activity on the cyclohexene hydrogenation. It was found that the charge transfer from Co to Ni and the alloying give rise to the refinement of CoNi grains and the improvement of its catalytic activity and stability. Thus, it may be possible to obtain better catalytic performance by tuning the metal/metal atomic ratio of bimetals.

## 1. Introduction

The catalytic hydrogenation of C=C double bonds is of vital importance in chemical industry [1,2,3,4,5,6]. Among various heterogeneous catalysts, noble metals, such as Ru [7,8], Rh [9,10], Pt [11,12,13], Au [14,15] and Pd [16,17], are most commonly used catalysts and exhibit superior catalytic performance on the unsaturated organic compound hydrogenation. Despite their outstanding performance, their inherent disadvantages, such as the high price, instability and limited resources, enormously limit their extensive application. Therefore, designing and synthesizing non-noble metal-based catalysts with excellent activity, selectivity and stability for the olefin hydrogenation is highly necessary. Non-noble metals, particularly those based on low-cost, non-noble and environmentally friendly bimetals, have attracted more and more interest, as they possess great potential in different application fields, such as catalysis [18,19,20], sensing [21,22], electrocatalysis [23,24] and so on.

Usually, alloying is an effective way to achieve grain refinement, so as to acquire more stable microstructures and more active sites. As seen in Ni-Cu [18,25], Co-Cu [19,26] and Ni-Co [20,24], different metal-metal interactions lead to enhanced catalytic activity. Liu et al. [18] reported the catalytic hydrogenation of furfural to tetrahydrofurfuryl alcohol over a pure Ni and bimetallic Ni-Cu catalysts, and the results showed that furfural on Ni-Cu bimetal exhibits a better conversion of 100% and a high selectivity of 90.3% to tetrahydorfurfuryl alcohol. Qiu et al. [19] found that the activation barrier for the catalytic hydrogenation of N_2_ to NH_3_ decreased with a Co-Cu catalyst for the Co and Cu synergistic effect. Corma et al. [20] studied the selective hydrogenation of nitroarenes to anilines in a CoNi@C catalyst, which exhibited a higher activity of about five time than the monometallic Co@C catalyst. In particular, they found that H_2_ chemisorption dissociation was the rate-determining step in the catalytic hydrogenation reaction. Indeed, the morphological tuning of bimetallic nanoparticles have been developed extensively.

The catalytic hydrogenation performance of a catalyst greatly depends on the efficiency of H_2_ dissociation and H atom migration. Ni-based catalysts have been widely studied for their relatively high activity in different catalytic hydrogenation reactions [19,20,27]. Yet, the stability is still a challenge for Ni-based catalysts and insight into the synergistic mechanism of metal-metal interaction remains insufficient. Co was found to have a well-structured stability [28,29], which might make it a good candidate to promote other active metals. Therefore, we introduce Co into Ni to generate a series of CoNi bimetals to perform a series of experiments and DFT calculations on the effects of Co in the catalytic hydrogenation performance of the CoNi bimetallic catalysts.

In the present study, we attempt to understand how Co influences the microstructure of CoNi bimetals and further affects the catalytic behavior of CoNi bimetallic catalysts. A series of Co*_x_*Ni*_y_* (*x* + *y* = 6) bimetallic catalysts were prepared by impregnation-reduction method, and cyclohexene was chosen as probe-molecule to study the promotion effect of Co on the catalytic olefin hydrogenation reactions. TG, H_2_-TPR, H_2_-TPD, XRD, BET/BJH, SEM, TEM and GC-MS were employed to characterize the physiochemical properties of the CoNi bimetallic catalysts. Meanwhile, DFT calculations were used to investigate the charge transfer between Co and Ni, and the transition state (TS) searched for hydrogen dissociation and migration. Our findings provide a valuable insight into metal-metal interactions for the improved catalytic performance on the catalytic hydrogenation and shed light on the design of non-precious metal-based bimetallic catalysts.

## 2. Experimental and Computational Details

### 2.1. Catalysts Preparation

A series of Co*_x_*Ni*_y_*/SiO_2_ (*x* + *y* = 6) catalysts with different Co/Ni atomic ratio (5:1, 4:2, 3:3, 2:4 and 1:5) and the same 5 wt.% metal loading were synthesized through a simple impregnation-reduction method using a commercial silica as support and aqueous solutions of Co(NO_3_)_3_·6H_2_O and Ni(NO_3_)_2_·6H_2_O as precursors, respectively. The synthesis was performed following the procedure of ref. [30] with some modifications. In the five parallel experiments, 1 g of SiO_2_ was placed in a 25 mL glass beaker, the Co and Ni precursor solution with different Co/Ni atomic ratios was added dropwise into the support. The mixtures were quickly stirred in a water bath at 60 °C for 12 h and then dried at 80 °C overnight. Subsequently, the solid mixtures were transferred to a tubular furnace and reduced under 500 °C in 10 vol.% H_2_/Ar atmosphere for 3 h. As a comparison, pure Co/SiO_2_ and pure Ni/SiO_2_ catalysts with the same metal loading were also prepared using the same method. The final samples were labeled as Co, Co_5_Ni_1_, Co_4_Ni_2_, Co_3_Ni_3_, Co_2_Ni_4_, Co_1_Ni_5_ and Ni, respectively. All chemicals were of analytical grade (Shanghai Chem. Co., Shanghai, China) and used without further purification.

### 2.2. Catalyst Characterization

The decomposition temperatures of the as-prepared samples were measured by a DTG-60H thermogravimetric (TG) analyzer. H_2_ temperature programmed reduction (H_2_-TPR) and H_2_ temperature programmed desorption (H_2_-TPD) were tested with an AutoChem-2950HP analyzer to analyze the reduction temperature difference and H_2_ chemisorption amount between the pure metals and the alloys. The phase structures of the as-prepared samples were tested by X-ray diffraction (XRD) utilizing a D8 Advance diffractometer with Cu-Ka radiation. To obtain the Brunauer–Emmett–Teller (BET) specific surface areas and the Barrett–Joyner–Halanda (BJH) pore size distribution and pore volume, nitrogen sorption experiments were performed with a Micromeritics ASAP 2020 HD88 sorption analyzer at 77 K. All samples were outgassed at 200 °C in vacuum for 4 h before measurement. The total pore volume was evaluated at the relative pressure of 0.995. The surface morphology of samples was analyzed using a JEOL JSM-6490LV scanning electron microscopic (SEM). Transmission electron microscope (TEM) pictures were obtained with a JEM-2100 apparatus. X-ray photoelectron spectroscopy (XPS) analysis was recorded on a VG Multilab 2000 spectrometer with Al-Ka radiation (1486.6 eV).

### 2.3. Catalytic Evaluation

All catalysts were evaluated through cyclohexene hydrogenation. The whole process was carried out in a 500 mL LABE high temperature and high pressure autoclave reactor, with a fixed stirring speed of 600 r/min. A total of 0.1 g of the catalyst and 1.0 g of the cyclohexene were dispersed using a 40 mL hexane solution. Before heating up, H_2_ was charged and discharged 3 times to exclude the air. The reactions were performed ranging from 25–100 °C at 2 MPa H_2_ pressure. An Agilent 8860-5977B gas chromatograph mass spectrometer (GC-MS) was used to analyze the reactant and products. The liquid sample was extracted with an interval of 15 min in the initial 60 min and of 30 min in the next 1 h.

### 2.4. Computational Methods

The density functional theory (DFT) calculations were performed within the generalized-gradient approximation (GGA), with the exchange-correlation functional of Perdew–Burke–Ernzerhof (PBE) [31,32]. The details are shown in the Appendix A.

## 3. Results and Discussion

### 3.1. H_2_-TPR, H_2_-TPD and XRD Analysis

In the present study, we attempted to understand how Co influences the catalytic performance of CoNi bimetallic catalysts with different Co/Ni atomic ratios. A series of Co*_x_*Ni*_y_*/SiO_2_ (*x* + *y* = 6) catalysts were prepared using the impregnation–reduction method. TG tests were carried out to distinguish the bimetallic from the monometallic precursors. Appendix A shows the TG curves of pure Ni, pure Co and Co_3_Ni_3_ bimetallic precursors. It can be seen that a mass loss of about 12.65 wt.% from room temperature to 200 °C was mainly attributed to the loss of crystal water. As the temperature rises, nickel nitrate and cobalt nitrate were further decomposed with a mass loss of about 5.56 wt.% to produce nickel and cobalt oxide with mixed valence. Compared to the pure Ni precursor, the bimetallic Co_3_Ni_3_ precursor shows a relatively faster decomposition during this stage, indicating the promoting effect in the existence of Co. The whole thermal decomposition was basically completed after 300 °C.

To further investigate the influence of Co in the CoNi bimetals, H_2_-TPR was proceeded on the Ni, Co and Co_3_Ni_3_ oxides from 50 °C to 750 °C. Figure 1a shows that both pure Co and pure Ni catalysts have only one main H_2_ consumption peak. located at 270 °C and 310 °C, respectively. Apparently, the metal oxides of both pure Ni and pure Co can be reduced at a relatively low temperatures. However, the bimetallic catalyst Co_3_Ni_3_ shows two H_2_ consumption peaks with obvious blue shifts to a lower temperature of 195 °C (Co) and 266 °C (Ni), suggesting that alloying can help to decrease the reduction temperature of bimetallic catalysts [33]. H_2_-TPD tests were also carried out to investigate the interactions between hydrogen and the active sites, as well as to confirm the amount of chemisorbed hydrogen, which can be used to estimate the metal dispersion. It can be seen from Table 1 that the amount of chemisorbed H_2_ on the CoNi catalyst is higher than on corresponding pure Co and Ni catalysts, suggesting that the number of CoNi active sites on the CoNi/SiO_2_ catalyst increased. This may be explained by the fact that the interaction between Co and Ni improved the distribution of CoNi on the surface of SiO_2_, agreeing with the results of H_2_-TPR.

Subsequently, the XRD analysis was also utilized to study the impact of the variation of Co/Ni atomic ratio to the bimetals. The results are shown in Figure 1b. It is found that all samples displayed two characteristic peaks located near 2θ = 44.4° and 51.7°. In particular, the main peaks of CoNi bimetals were between pure Ni and pure Co, according to PDF cards JCDPS#04-0850 and JCDPS#01-1254, indicating the appearance of (111) and (200) planes of CoNi alloy, respectively [34]. No metallic oxides were detected, suggesting that CoNi bimetals can be readily reduced under the present conditions.

### 3.2. BET and BJH Analysis

To further investigate the texture properties of different CoNi bimetals, N_2_ sorption isotherms of the as-prepared 5 wt.% Co*_x_*Ni*_y_*/SiO_2_ (*x* + *y* = 6), 5 wt.% Co/SiO_2_ and 5 wt.% Ni/SiO_2_ catalysts are shown in Figure 2a. Each curve shows a H2 type hysteresis loop, which is distributed in the relative pressure range of P/P_0_ 0.5—0.9. This is a typical type IV curve, which means that these samples possess mesoporous structures. Moreover, obvious adsorption platforms can also be seen from the curves, indicating the relatively regular pore structure of the samples. Figure 2b shows distinct hierarchical pore structure characteristics of Co*_x_*Ni*_y_*/SiO_2_ bimetallic catalysts, which are mainly distributed at about 6.6 nm and 10.5 nm, respectively. Clearly, this pore size distribution is very favorable for the formation of small metallic nanoparticles and is highly suitable for the diffusion of small reactant molecules.

The texture parameters of the as-prepared samples are listed in Table 2. After loading the bimetals, the S_BET_ of the samples decreased to about 300 m^2^/g. The average pore size increased slightly and the pore volume decreased partially, which may be caused by the entry of CoNi nanoparticles into SiO_2_ pores. In addition, the grain size of CoNi nanoparticles was calculated according to Scherrer formula. It can be seen from the table that the grain size of CoNi alloy is distributed between 11.89 nm and 13.19 nm, which is much smaller than that of pure Ni (23.65 nm) and a little bigger than pure Co (10.42 nm). As a result of this, one can conclude that the existence of Co can somehow help to inhibit the growth of the CoNi nanoparticles.

### 3.3. SEM and TEM Analysis

SEM and EDS mapping tests were also carried out on Co_3_Ni_3_ in ofer to understand the micro-morphology information of the CoNi bimetallic catalyst. Figure 3a–c shows the SEM images of Co_3_Ni_3_ with different resolutions, it was found that SiO_2_ had a porous structure. The EDS mapping of Co_3_Ni_3_/SiO_2_ (Figure 3d–g) proved that the elements of Co and Ni are uniformly distributed, indicating that the Co_3_Ni_3_ bimetallic nanoparticles can be supported stably on the surface and pores of SiO_2_.

TEM analysis was also carried out to investigate the morphology and size of the reduced Co, Ni and Co_3_Ni_3_ catalysts supported on SiO_2_. Figure 4a–c shows the structure of the Co/SiO_2,_ Ni/SiO_2_ and Co_3_Ni_3_/SiO_2_ samples, respectively. Co nanoparticles are evenly dispersed on the SiO_2_ surface (Figure 4a). The histogram of the average particle size distribution reveals that the Co has the smallest particle size distribution of 8.31 nm, while the particle size of pure Ni is distinctly larger than both pure Co and Co_3_Ni_3_. The widely distributed nanoparticles as well as the biggest average particle size of 44.2 nm imply that pure Ni nanoparticles are prone to aggregation (Figure 4b). However, Co_3_Ni_3_ nanoparticles (Figure 4c) show an average particle size of only 12.17 nm, far below the average particle size of pure Ni and close to that of pure Co. This trend is consistent with the results from XRD, which further proves that the existence of Co in the bimetallic alloy effectively inhibits the growth and agglomeration of CoNi grains and improves the stability of alloy nanoparticles. Clearly, it is possible to confirm that CoNi bimetal is much more stable than pure Ni or Co, implying that the existence of the strong interaction between Co and Ni [30,35]. In addition, the HRTEM result shown in Figure 4d reveals that the lattice spacing of the (111) and (200) planes of Co_3_Ni_3_ bimetal are 0.206 nm and 0.172 nm, respectively, which is between pure Co and pure Ni, suggesting the formation of the CoNi alloy [34].

### 3.4. XPS Analysis

XPS was used to analyze the elemental valence state differences between pure Co/SiO_2_, pure Ni/SiO_2_ and Co_3_Ni_3_/SiO_2_ catalysts. Appendix A reveals the full spectral O 1s, the Co 2p and the Ni 2p of all catalysts. The Co 2p spectra (Figure 5a) of both pure Co/SiO_2_ and Co_3_Ni_3_/SiO_2_ show two main peaks at around 781.2 eV (Co 2p_3/2_) and 797.8 eV (Co 2p_1/2_), which correspond to the Co^2+^ ion. It was noted that two peaks located at 778.8 eV and 793.6 eV that belong to Co^0^ can only be found in Co_3_Ni_3_/SiO_2_, indicating its relatively better anti-oxidative ability. The main peak displacement of 0.2 eV from high binding energy to low binding energy in Co 2p_3/2_ also reveals the fact that more Co low-valence ions exist on the surface of Co_3_Ni_3_/SiO_2_ than on pure Co/SiO_2_ [36]. Figure 5b shows the Ni 2p spectrum of pure Ni/SiO_2_ and Co_3_Ni_3_/SiO_2_, and both catalysts exhibit the characteristic peaks of Ni^2+^ and Ni^0^ near 855.58 eV and 852.18 eV for 2p 3/2 and 872.88 eV and 870.58 eV for 2p 1/2, respectively [36,37]. The main peak of Ni^2+^ and Ni^0^ shifts from lower binding energy to high binding energy, implying that Co_3_Ni_3_/SiO_2_ possesses more Ni^2+^ and less Ni^0^ than pure Ni/SiO_2_. This can be attributed to the fact that the particle size of pure Ni is much larger than that of Co_3_Ni_3_, resulting in the relatively stable superficial Ni valence.

### 3.5. Catalytic Hydrogenation Performance

In order to investigate the influence of different atomic ratios of Co/Ni on the catalytic olefin hydrogenation performance, cyclohexene was chosen as the probe molecule and a series of hydrogenation experiments with different Co*_x_*Ni*_y_* catalysts were carried out under the setting conditions of 100 °C, at a H_2_ pressure of 2 MPa for 3 h. The conversion curves vs. time are shown in Figure 6, while the conversion results of cyclohexene hydrogenation after 3 h can be seen in Table 3. Co_3_Ni_3_ exhibits a superior catalytic activity to the other catalysts, realizing a 100% hydrogenation conversion from cyclohexene to cyclohexane within 90 min (Figure 6a). The lowest conversion of 62.4% for pure Co/SiO_2_ catalyst may suggest that the main active site is Ni for the CoNi bimetals. Moreover, the corresponding histogram of the hydrogenation reaction within 60 min is also shown in Figure 6b. It can be seen that the catalytic activity of CoNi bimetals shows an upward trend as the atomic ratio of Co/Ni increases to 3:3 and subsequently decreases as the content of Co further rises. This trend can be explained by the fact that the reactivity of metal surface increases with the enhanced Co-Ni interaction. While further rise of Co content will lead to the increase of high-valence inactive metal ions. This result is consistent with the above TEM and XPS conclusions, implying that optimal catalytic performances can be obtained by the appropriate tuning the atomic ratio of Co/Ni.

In addition, to further understand the influence of reaction temperature and H_2_ pressure on the catalytic performance of Co_3_Ni_3_, the temperature gradient experiments of cyclohexene hydrogenation at the reaction condition of 25 °C to 100 °C, at a 2 MPa H_2_ pressure and 6 h, were proceeded. As is shown in Figure 7a, the conversion of cyclohexene is only 55.44% at 25 °C and can achieve 100% at 60 °C within 6 h. Clearly, the conversion of cyclohexene hydrogenation increases as the temperature rises. Upon the temperature reaching 80 °C, a 100% conversion can be realized within 4 h. Subsequently, the H_2_ pressure gradient experiments were carried out at 0.1 MPa to 3 MPa, at 100 °C, also within 90 min. Figure 7b shows that the catalytic hydrogenation conversion of cyclohexene increases from 74.5% to 100% as the H_2_ pressure varies from 0.1 MPa to 1 MPa. When further prolonging the reaction time to 180 min, the conversion can also reach 100% even under the lowest H_2_ pressure of 0.1 MPa, indicating that the catalytic hydrogenation of cyclohexene is more sensitive to temperature than the H_2_ pressure. These results are superior to certain noble metal catalysts reported by previous research [38,39,40], as shown in Table 4. In addition, considering the poor stability of the monometallic Ni catalyst, in which the metal nanoparticles readily agglomerate [41,42], the stability of Co_3_Ni_3_ bimetallic was also verified through cyclic catalytic hydrogenation tests at 100 °C and 2 MPa H_2_ pressure within 90 min. Appendix A shows the conversion after three hydrogenation cycles and only a slight activity decay was detected, indicating the good stability of bimetallic Co_3_Ni_3_. These results proved that bimetallic Co_3_Ni_3_ may be a potential catalyst for the catalytic hydrogenation of unsaturated organic molecules.

### 3.6. DFT Calculation

DFT calculations were also employed to further explore the promotion effect of Co in the catalytic properties of CoNi bimetallic catalysts. Despite the high complexity of the properties and microstructures of a true CoNi nanoparticle, the CoNi clusters used here are at a sub-nano level. Previous research has already confirmed that when a sub-nano cluster consists or exceeds six atoms, it may systematically and adequately illustrate the entire catalytic reaction procedure to obtain useful insights into different metal-metal interactions [43,44,45,46]. Thus, Co*_x_*Ni*_y_* clusters (*x* + *y* = 6) were chosen as the subsequent research models.

Usually, the catalytic performance of a cluster is remarkably affected by its stability and charge distribution. Therefore, a series of Co*_x_*Ni*_y_* clusters with the lowest energy structures were obtained by massive structural optimizations. The optimized lowest energy structures of Co*_x_*Ni*_y_* clusters, the Mulliken charge distribution as well as the homologous average formation energies are shown in Figure 8a. It seems that the formation energies of Co*_x_*Ni*_y_* clusters increase with the rise of Co content. Therefore, with the existence of Co, Co*_x_*Ni*_y_* bimetallic clusters become more steady than corresponding pure Ni_6_ clusters. Meanwhile, Mulliken charge analysis (Figure 8a) suggests that the Ni atoms have higher charge density than the Co atoms in Co*_x_*Ni*_y_* bimetals due to the charge transfer from Co to the adjacent Ni atoms. This result agrees well with the fact that the electro-negativity of Ni (1.91) is a little higher than that of Co (1.88). Notably, atoms of the same type that are located at the opposite positions carry the same number of charges. As the Co atom number rises, the number of negative charges carried by each Ni increase gradually to the highest value of −0.017 eV (Co_3_Ni_3_) and subsequently decreases as the number of Co atoms surpass the number of Ni atoms. Furthermore, the maximum hydrogen capacity of Co*_x_*Ni*_y_* clusters was also calculated, and Co_3_Ni_3_ and Co_2_Ni_4_ show the highest H capacity of 16 (Appendix A), while the AIMD results also proved that Co_3_Ni_3_ has the ability to hold 16 H atoms (Appendix A). These results agree well with the above H_2_-TPD analysis as well as the previous report that the charge transfer from support or assistant to active metal may lead to a promotion in the catalytic activity and H capacity [47].

H atoms usually present high mobility on metal clusters or surface, as they can readily migrate to the adjacent adsorption sites with a relatively low energy barrier. As a result of this, the chemisorption structure of the H_2_ molecule on Co*_x_*Ni*_y_* is always unstable. The detailed process of H_2_ molecule dissociation and H atoms migration on the Co_3_Ni_3_ cluster is shown in Figure 8b. Initially, an H2 molecule is adsorbed on the top site of Ni (R1) and is dissociated by the Ni atom with a low transition state energy barrier of 0.11 eV (TS1). Surprisingly, this value is superior to the corresponding Ni_6_, Pt_6_ (0.6~1.0 eV) [46] and Pd_6_ clusters (0.37~0.4 eV) [48,49] and is very close to the Pt (111) surface (~0.03 eV) [50]. Subsequently, the two H atoms move to their adjacent edges, which is considered to be the most stable structure for the Co_3_Ni_3_ octahedral cluster. Nevertheless, the further migration of the H atom (e.g., H atom move from P1 to P2) only needs a low energy barrier of 0.24 eV (TS2) implying that H_2_ molecule dissociation and H atoms migration processes are mild on the Co_3_Ni_3_ cluster. A slight increase in the energy barrier (0.26 eV) is found in the process of TS3 (Figure 8b). In a word, the low H_2_ dissociative energy barrier and the feasible reaction energies as well as the favorable migration barrier all suggest that these processes on Co_3_Ni_3_ bimetal are thermodynamically controlled.

## 4. Conclusions

In this paper, the promotion effect of Co on the physicochemical properties and the catalytic hydrogenation performance of CoNi bimetallic catalysts were studied by both experiments and DFT calculations. It was found that Co_3_N_3_ shows the superior catalytic activity over the other catalysts in olefin catalytic hydrogenation, and a 100% conversion of cyclohexene to cyclohexane can be achieved under mild conditions (100 °C, 1 bar) in 3 h. The presence of Co can effectively decrease the reduction temperature of bimetallic precursors and inhibit the growth and agglomeration of CoNi bimetallic nanoparticles, making the CoNi catalysts more stable and active. DFT calculations showed that the charge transfer from Co to Ni may result in the increase in microstructure stability and the improvement of the catalytic performance of CoNi bimetals. In particular, the H_2_ dissociation activation energy on Co_3_N_3_ is only 0.11 eV, lower than corresponding pure Ni_6_, Pt_6_ and Pd_6_ clusters. Our findings reveal that the existence of Co can effectively promote the physicochemical properties of CoNi bimetals and the appropriate Co/Ni atomic ratio will lead to the excellent catalytic hydrogenation performance of Ni sites. We believe that these results may provide useful insights for tuning the ingredients and metal-metal interactions in the design and synthesis of non-noble bimetallic catalysts.

## Figures and Tables

**Figure 1 nanomaterials-13-01939-f001:**
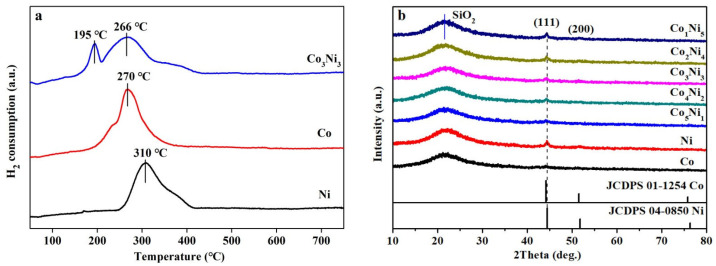
(**a**) H_2_-TPR tests of Ni, Co and Co_3_Ni_3_ oxides and (**b**) the XRD patterns of pure Co, pure Ni and CoNi bimetals with different Co/Ni atomic ratios.

**Figure 2 nanomaterials-13-01939-f002:**
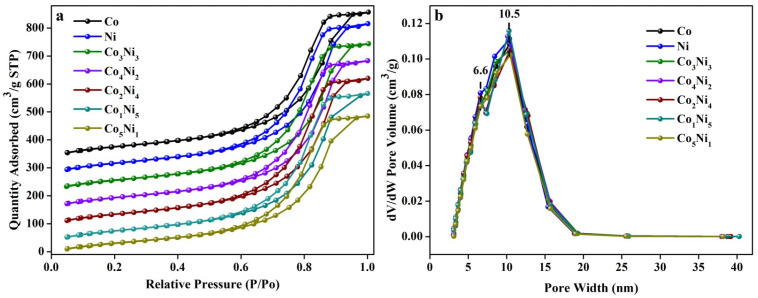
(**a**) N_2_ adsorption/desorption isotherms of Co*_x_*Ni*_y_*/SiO_2_, pure Co/SiO_2_ and pure Ni/SiO_2_ samples and (**b**) their corresponding BJH pore size distributions.

**Figure 3 nanomaterials-13-01939-f003:**
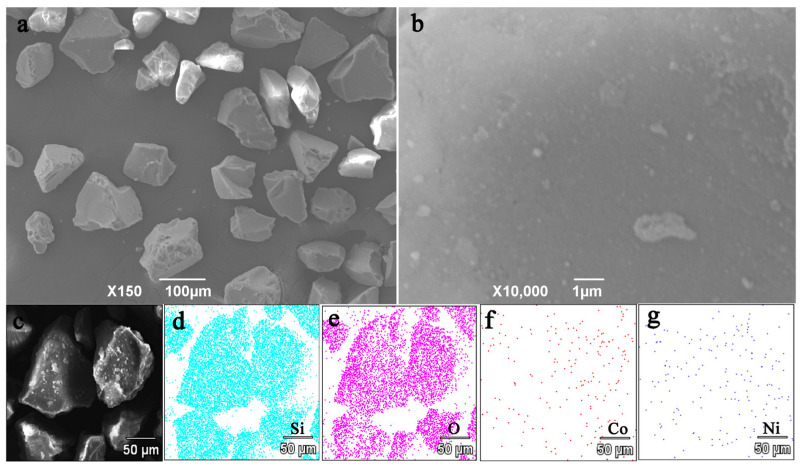
SEM images of (**a**–**c**) Co_3_Ni_3_ with different resolutions, and (**d**–**g**) the EDS mapping of the Si, O, Co and Ni elements.

**Figure 4 nanomaterials-13-01939-f004:**
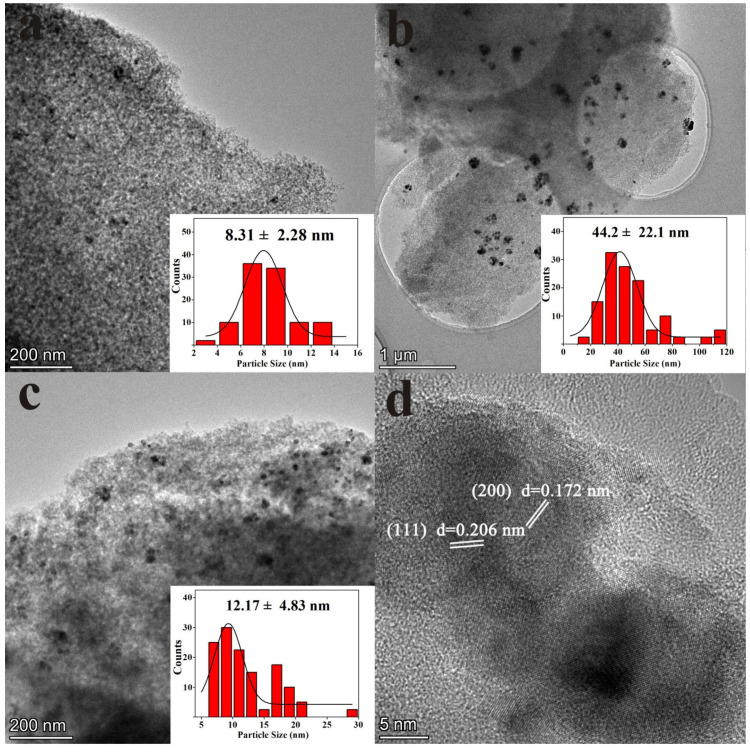
TEM images and particle size distribution of (**a**) pure Co/SiO_2_, (**b**) pure Ni/SiO_2_, (**c**) Co_3_Ni_3_/SiO_2_ bimetal and (**d**) HRTEM image of Co_3_Ni_3_/SiO_2_ bimetal.

**Figure 5 nanomaterials-13-01939-f005:**
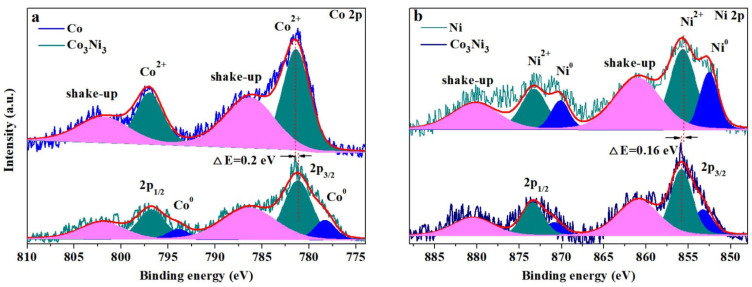
XPS spectra of pure Co/SiO_2_, pure Ni/SiO_2_ and Co_3_Ni_3_/SiO_2_; (**a**) Co 2p spectral analysis and (**b**) Ni 2p spectral analysis.

**Figure 6 nanomaterials-13-01939-f006:**
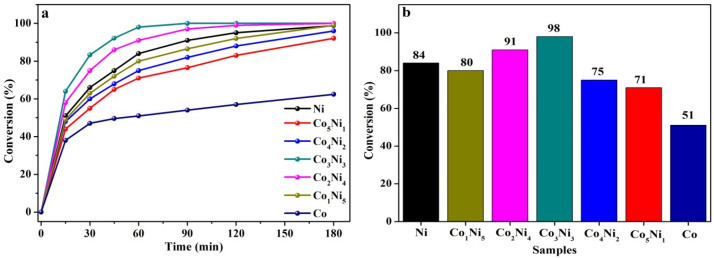
Hydrogenation conversion curves of cyclohexene over (**a**) different Co*_x_*Ni*_y_* catalysts and (**b**) their corresponding histogram within a period of 60 min.

**Figure 7 nanomaterials-13-01939-f007:**
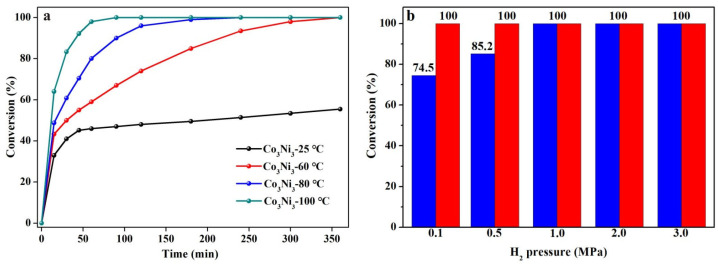
Curves of catalytic hydrogenation conversion of cyclohexene on Co_3_Ni_3_ (**a**) at 25–100 °C and 2 MPa H_2_ pressure within 360 min for temperature gradient experiments and (**b**) at 0.1–3 MPa and 100 °C for H_2_ pressure gradient experiments. Blue bar: 90 min; red bar: 180 min.

**Figure 8 nanomaterials-13-01939-f008:**
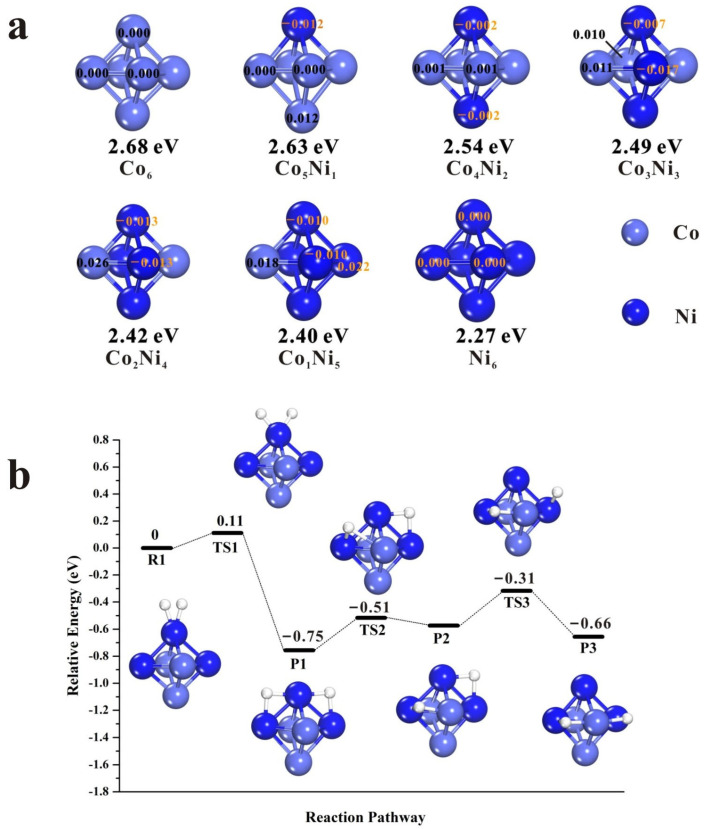
(**a**) The optimized most stable structures of Co*_x_*Ni*_y_* clusters together with Mulliken charge distribution and the corresponding average formation energies and (**b**) the calculated reaction pathway of H_2_ dissociation and the subsequent H atoms migration on the Co_3_Ni_3_ octahedral cluster. Purple ball: Co atoms; blue ball: Ni atoms; white ball: H atoms.

**Table 1 nanomaterials-13-01939-t001:** H_2_-TPD data of Co/SiO_2,_ Ni/SiO_2_ and Co_3_Ni_3_/SiO_2_ catalysts.

Samples	H_2_ Desorbed (umol/g)	Dispersion (%)
5 wt% Co/SiO_2_	9.8	10.6
5 wt% Ni/SiO_2_	5.1	5.8
5 wt% Co_3_Ni_3_/SiO_2_	11.8	10.2

**Table 2 nanomaterials-13-01939-t002:** Texture parameters of Co*_x_*Ni*_y_*/SiO_2_ (*x* + *y* = 6) bimetallic catalysts and SiO_2_ support.

Samples	S_BET_ (m^2^/g)	Average Pore Size (nm) ^a^	V_tot_ (cm^3^/g)	Grain Size (nm) ^b^
SiO_2_	322.74	8.64	0.95	-
5 wt% Co/SiO_2_	299.52	8.73	0.87	10.42
5 wt% Ni/SiO_2_	306.10	8.64	0.89	23.65
5 wt% Co_5_Ni_1_/SiO_2_	280.96	8.69	0.82	13.19
5 wt% Co_4_Ni_2_/SiO_2_	294.60	8.87	0.88	12.18
5 wt% Co_3_Ni_3_/SiO_2_	299.14	8.81	0.88	12.06
5 wt% Co_2_Ni_4_/SiO_2_	298.44	8.70	0.87	11.89
5 wt% Co_1_Ni_5_/SiO_2_	298.87	8.79	0.88	11.98

^a^ the average pore size distribution was calculated using the BJH method. ^b^ the grain size of each metal and bimetal nanoparticle was calculated using the Scherrer formula.

**Table 3 nanomaterials-13-01939-t003:** Catalytic activity of 5 wt% Co*_x_*Ni*_y_*/SiO_2_ (*x* + *y* = 6) catalysts for the hydrogenation of cyclohexene under 100 °C, at a 2 MPa H_2_ pressure for 3 h.

Catalysts	Temperature (°C)	H_2_ Pressure (MPa)	Time (min)	Conversion (%)
5 wt% Co/SiO_2_	100	2	180	62.4
5 wt% Ni/SiO_2_	100	2	180	98.8
5 wt% Co_5_Ni_1_/SiO_2_	100	2	180	92.1
5 wt% Co_4_Ni_2_/SiO_2_	100	2	180	96.0
5 wt% Co_3_Ni_3_/SiO_2_	100	2	90	100
5 wt% Co_2_Ni_4_/SiO_2_	100	2	180	100
5 wt% Co_1_Ni_5_/SiO_2_	100	2	180	99.0

**Table 4 nanomaterials-13-01939-t004:** Comparison of 5 wt.% Co_3_Ni_3_/SiO_2_ bimetallic catalysts with other reported catalysts for the hydrogenation of cyclohexene under different conditions.

Catalysts	Temperature (°C)	H_2_ Pressure (MPa)	Time (min)	Conversion (%)
5 wt% Co_3_Ni_3_/SiO_2_	25	2	360	55.5
5 wt% Co_3_Ni_3_/SiO_2_	60	2	360	100
5 wt% Co_3_Ni_3_/SiO_2_	80	2	240	100
5 wt% Co_3_Ni_3_/SiO_2_	100	0.1	90	74.5
5 wt% Co_3_Ni_3_/SiO_2_	100	0.5	90	85.2
5 wt% Co_3_Ni_3_/SiO_2_	100	1	90	100
5 wt% Co_3_Ni_3_/SiO_2_	100	2	90	100
5 wt% Co_3_Ni_3_/SiO_2_	100	3	90	100
2 wt% Pt/TiO_2_-P [38]	20	3	170	19.45
5 wt% Fe_2_Co_4_/SiO_2_ [30]	100	2	300	100
0.1 wt% Pd/MoO_3_ [39]	150	3	270	100
Commercial 5 wt% Ru/Al_2_O_3_ [40]	150	3	180	100

## Data Availability

All data are contained within the article.

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
