# Peer review of "Co-Promoted CoNi Bimetallic Nanocatalyst for the Highly Efficient Catalytic Hydrogenation of Olefins"

_nanomaterials, 2023, doi:10.3390/nano13131939_

Round 1

Reviewer 1 Report

Paper is focused to demonstrate that the incorporation of Co in Ni/SiO2 contribute to enhance the catalytic performance in cyclohexene hydrogenation.

Unfortunately the English is poor and the conclusion drawn are not sustained by significant scientific evidences.

Questions:

-Why the authors chosen the cyclohexane hydrogenation? Probably other more specific hydrogenation reactions they could be chosen.

-The formation of CoNi alloy which should be the key to demonstrate the difference in catalytic performance is not really demonstrated. TEM images used to demonstrate the formation of alloy are not reliable.

-The evaluation of active metallic sites is fundamental. H2 TPD could help to understand why Co3Ni3 is more active than other ones. In fact the difference in activity could be linked to the H2 adsorption capacity.

Round 2

Reviewer 1 Report

Not particular comments on revised version.